# Introduction of Complementary Foods and the Risk of Sensitization and Allergy in Children up to Three Years of Age

**DOI:** 10.3390/nu15092054

**Published:** 2023-04-24

**Authors:** Magdalena Chęsy, Aneta Krogulska

**Affiliations:** Department of Pediatrics, Allergology and Gastroenterology, Ludwik Rydygier Collegium Medicum in Bydgoszcz, Nicolaus Copernicus University in Torun, ul. M. Curie Skłodowskiej 9, 85-094 Bydgoszcz, Poland; aneta.krogulska@cm.umk.pl

**Keywords:** allergy, complementary foods, children, nutrition, sensitization

## Abstract

Background: Allergy is known to be influenced by both diet and the immune system. In addition, the time of first exposure to food allergens and their type appear to play a particularly important role in the development of allergies. Aim: To determine the influence of the time of exposure, and the type, degree of processing, and frequency of supply of complementary foods and the development of sensitization and allergies in children up to three years of age. Materials and metods: The study was conducted prospectively in two stages. The first stage included 106 children aged 6–18 months, while the second stage included 86 children selected from the first stage, after a further 12 months. A questionnaire based on validated FFQ sheets was created for the purpose of the study. The following were assessed: nutrition in the first year of life (time, type, degree of processing), frequency of supply of complementary foods and allergic symptoms, sIgE concentration against 10 foods and 10 inhalant allergens. Four groups of patients were formed. This paper presents the results of the second stage of the study. Results: For all participants, allergenic products, viz. hen’s egg, milk, peanuts, wheat, soybean, fish, tree nuts and shellfish were typically introduced at an age of 7 to 12 months. During this period, egg white was introduced in 47 (85.5%) children with allergy (*p* = 0.894), in 29 (82.9%) with allergy and sensitization (*p* = 1.00), and in 38 (82.6%) children with sensitization alone (*p* = 0.533). Milk was introduced at 7 to 12 months in 35 (64.8%) children with allergy (*p* = 0.64), 22 (64.7%) with both allergy and sensitization (*p* = 0.815), and 26 (57.8%) children with sensitization alone (*p* = 0.627). For other foods, the time of introduction appeared not to significantly influence the presence of allergies or sensitization. Heat-treated peanuts were introduced significantly more often to children without allergies and without sensitization (*n* = 9; 56.2%) than those without allergies but with sensitization (*n* = 6; 54.5%) (*p* = 0.028). Fish was consumed significantly more often by children with allergies, i.e., 1–3x/week (*n* = 43; 79.6%) than children without allergies, i.e., 1–3x/month (*n* = 9; 30%) (*p* = 0.009). Conclusions: No relationship was observed between the introduction time of complementary foods, including allergenic ones, or their type, and the development of allergies and sensitization in children up to three years of age. The degree of processing and the frequency of supply of products may affect the development of allergies and sensitization.

## 1. Introduction

Allergic diseases constitute a significant public health problem affecting approximately 30% of the global population [1]. The role played by dietary factors in the promotion of immunotolerance and in the prevention of allergic diseases has been increasingly emphasized by recent studies. Evidence indicates that the development of allergy may be influenced by the relationship between diet, microbiota, and the immune system. It has been shown that diet affects the quality of microbiota colonizing the digestive tract [2]. Dietary factors are known to have an immunomodulatory effect on the functioning of the immune system, both indirectly, through the microbiota, and directly, by increasing cell activity and activating immunoglobulin production [3]. It has been demonstrated that the diet affects the immune system via acquired immunity (e.g., by increasing the production of antibodies, development, and increasing the activity of Treg cells) and innate immunity (e.g., by increasing the activity of Natural Killer cells and phagocytes) [4]. In such cases, allergy development seems to be primarily influenced by the time of first exposure to dietary components and their type [5].

Environmental factors are believed to exert their greatest impact on the functioning of the immune system during the prenatal period and infancy [5]. It was assumed that food should not be introduced into the child’s diet earlier than 17 weeks, nor later than 26 weeks. Currently, international and Polish scientific societies recommend exclusive breastfeeding for the first four to six months, and that complementary foods should be introduced when the child is between four and six months of age; in addition, there are no special recommendations regarding allergenic foods and no dietary restrictions are recommended for children after four to six months [6,7,8]. In infants at high risk of developing allergies, complementary foods should be introduced as in healthy children, taking into account family and cultural preferences and being aware of the possible occurrence of disturbing symptoms after food consumption [9].

The basis of infant nutrition comprises milk and complementary foods, with the latter being solid or liquid foods other than natural or synthetic milk given during weaning. Both the time of introducing complementary foods, as well as their type, appear to have an impact on the psychomotor and physical development of a child. These factors are also important in the development of diet-related diseases and taste preferences, as well as the sense of taste and allergies [10].

Studies evaluating the relationship between the timing of infant food introduction and the development of allergies are conflicting; some do not find any justification for early introduction of complementary foods, while others indicate that this may have a protective effect against developing allergies [11,12,13]. It is hence necessary to clarify whether, and which, components of the diet can modulate the development of allergic diseases, and whether the time of their introduction plays a role.

Therefore, the aim of the study was to determine whether the time of introduction, type, degree of processing, and the frequency of supply of introduced complementary foods influence the development of sensitization and allergy in children under three years of age.

## 2. Materials and Methods

This prospective study was performed in a group of 136 children aged 6–18 months. All participants were patients of the Department of Paediatrics, Allergology and Gastrenterology, Collegium Medicum NCU, Allergy Clinic, and Communal Nursery in Łochowo (Kujawsko-Pomorskie Voivodeship) in the period from 1 September 2016 to 30 December 2019.

The study was conducted in two stages. In total, 106 children between the ages of 6 and 18 months were eligible for Stage I. Of these, 86 were selected to continue to Stage II after a further 12-month period. The inclusion criteria for the study were ages 6 to 18 months and consent from the legal guardian, while the exclusion criteria comprised the presence of any severe chronic diseases that could affect the study outcome.

The study used a questionnaire based on the validated GA2LEN Food Frequency Questionnaire [14] and the NHANES Food Frequency Questionnaire [15], as modified by the author. The questionnaire was forwarded to the children’s legal guardians after they were qualified for the study. Nutrition in the first year of life was recorded, specifying the time of introduction of the supplementary foods (≤3 months; 4–6 months; 7–12 months), as well as their type, degree of processing, and frequency.

In Stages I and II, the children were clinically assessed by an allergist. In addition, antigen-specific IgE antibodies (sIgE) were determined in each child against 10 food allergens (peanut, milk, egg white, egg yolk, potato, carrot, cod, apple, soybean, wheat flour) and 10 inhalant allergens (birch pollen, timothy grass, mugwort pollen, D. pteronyssinus, D. farinae, dog epidermis, cat epidermis, horse epidermis, Aspergillus fumigatus, Cladosporium herbarium). In order to assess asIgE, blood was collected at two time points: in the first stage in children aged 12–18 months, and in the second stage after another 12–18 months. The sIgE were identified based on immunoenzymatic testing with the Polycheck kit (BioCheck GmbH, Leipzig, Germany). Children with sIgE ≥ 0.35 kU/L were considered sensitize.

Children with a clear history (i.e., children with anaphylaxis or anaphylactic reaction after ingestion of a specific allergen) and confirmed sensitization to a given food allergen were not challenged. Oral provocation tests were performed on allergic children with no clear medical history. In contrast, children with non-IgE-independent milk or egg allergy were put on a diagnostic elimination diet and then re-exposed to the allergen in the diet. In the case of improvement on the elimination diet, and then deterioration after the inclusion of the allergen in the diet, food allergy was confirmed. Based on the obtained data, clinical symptoms and sIgE results, the examined children were divided into four subgroups. The present study discusses only the results of Stage II of the study, in which the children were divided into four groups based on the presence of allergy and/or sensitization:-Group A: children with allergic symptoms and concomitant sensitization (i.e., children with IgE-mediated allergy);-Group B: children without allergic symptoms, with concomitant sensitization sensitized children);-Group C: children with allergic symptoms, without concomitant sensitization (i.e., children with non-IgE-mediated allergy);-Group D: children without allergic symptoms and without concomitant sensitization (i.e., healthy children).

The study group consisted of children from groups A, B, and C, while the control group consisted of children from group D.

The characteristics of the study group are presented in Table 1, and the study design in Figure 1.

### 2.1. Statistical Analysis

The results were analyzed using the *t*-test, Mann-Whitney-Wilcoxon test, chi-square test, Fisher’s test and logistic regression where appropriate. All calculations were performed using the R package, version 3.0.3. Statistical significance was assumed for *p* < 0.05.

### 2.2. Ethical Considerations

The study was approved by the Bioethical Committee of the Collegium Medicum in Bydgoszcz, Nicolaus Copernicus University in Toruń (KB 239/2016).

## 3. Results

Allergic symptoms were observed together with sensitization in 35 children (40.6%) (Group A), allergic symptoms without sensitization in 20 children (23.3%) (Group C), and sensitization alone without allergic symptoms in 11 (12.8%) (Group B). In addition, 20 (23.3%) children reported no allergic symptoms and no sensitization (Group D). Among 55 (100%) children with allergies (Groups A + C), atopic dermatitis (AD) was diagnosed in 17 (30.9%), egg allergy in 15 (27.3%), cow’s milk protein allergy in 11 (20%), peanut allergy in two (3.6%), asthma in eight (14.6%) and allergic rhinitis (AR) in two (3.6%).

Regarding the time of introduction of allergenic foods, i.e., hen’s egg, milk, peanuts, wheat, soybeans, fish, tree nuts, and shellfish, most were introduced at 7 to 12 months of age, regardless of the study group (Table 2). During this period, egg white was introduced in 47 (85.5%) allergic children (Groups A + C) and in 27 (87.1%) non-allergic children (Groups B + D) (*p* = 0.894). It was introduced in the period 7–12 months in 29 (82.9%) children with allergies and sensitization (Group A) and in 18 (90%) children without allergies or sensitization (Group D) (*p* = 1.00); it was also introduced during this period in 38 (82.6%) children in the allergic group (Groups A + B) and 36 (90%) in the non-allergic group (Groups C + D) (*p* = 0.533).

In this time, milk was introduced in 35 (64.8%) children with allergy and in 18 (60%) without allergy (*p* = 0.64), as well as in 22 (64.7%) with allergy and sensitization and in 14 (73.7%) without allergy or sensitization (*p* = 0.815). It was introduced in 26 (57.8%) children with sensitivity and in 27 (69.2%) without (*p* = 0.627).

As in the case of egg white and milk, none of the other complementary foods demonstrated any significant differences in the timing of their introduction into the diet with regard to the presence of allergies or sensitization (Table 2).

Regarding the degree of food processing, only the processing of peanuts appeared to have a significant influence on the development of allergies or sensitivity. The cooked form was chosen in nine (56.2%) children in the group without allergic symptoms or sensitivity; this value is significantly higher than in the group with both allergic symptoms and sensitivity (*n* = 3; 27.3%), where the raw form (peanut flour) was dominant (*n* = 6, 54.5%) (*p* = 0.028) (Figure 2).

Regarding the influence of the frequency of food supply, it was found that children with symptoms of allergy (*n* = 43; 79.6%) were given fish significantly more often, usually one to three times a week, than children without allergic symptoms (*n* = 9; 30%) who received fish less than one to three times per month (*p* = 0.009) (Figure 3).

The frequency of supply and the degree of food processing of other foods did not differ significantly between the study groups (Table 3).

It was also shown that diet diversification during the first year of life has a crucial relationship with allergy and/or sensitization occurrence. It was verified that, during the third month of life, a meaningfully larger number of parents (*n* = 15; 27.3%) of children with allergy symptoms introduced to their offspring’s diet one to two groups of food products in comparison with parents of children without allergy symptoms (*n* = 2; 6.5%) (*p* = 0.024). Moreover, children with allergy and/or sensitization were introduced to a smaller number of food product groups in the 6th month of life in comparison with children without allergy and/or sensitization (*p* < 0.003; *p* < 0.001; *p* = 0.008); analogously in the 12th month (*p* = 0.001; *p* < 0.001; *p* = 0.001).

The logistic regression was performed to identify independent factors associated with the occurrence of allergies. Table 4 presents the results of the univariate model (Model I) and of the multivariate model (Model II), comparing children from Groups A + C to those from Groups B + D. The conducted analysis showed a significant relationship between the diagnosis of allergies in the examined children and the occurrence of AD in their families. The risk of allergy was 17 times higher in children with a family history of AD than in those without AD in the family (aOR = 17.6; 95%CI 1.9–159.6; *p* = 0.011).

Another independent factor associated with the occurrence of allergies in the examined children was feeding with the use of ready-made products—their frequent use in the diet increased the risk of allergies by 11 times (aOR = 11.5; 95%CI 2.7–49.7; *p* = 0.001). However, the earlier introduction of a varied diet significantly reduced the risk of allergies. The introduction of more product groups into the child’s diet at 6 months of age (5–6 vs. 0–4 or 7 vs. 0–4) reduced the risk of allergies; aOR = 0.17 (95%CI 0.04–0.71) (*p* = 0.015) and aOR = 0.08 (95%CI 0.01–0.58) (*p* = 0.012), respectively. Additionally, a more varied diet at 12 months is associated with a lower risk of allergy; aOR = 0.14 (95%CI 0.03–0.57) (Table 4).

When comparing children from Group A to those from Group D, the occurrence of allergies and sensitization was found to be significantly associated with various factors, including the presence of food allergy, inhalant allergy or AD in the family, and a period of exclusive breastfeeding and diet diversity (Table 5). The logistic regression analysis found a significant relationship between the occurrence of allergies and sensitization in the examined children and a diagnosis of inhalation allergy in their families: more specifically, the risk of allergy and sensitization was five times higher in children with a family history of inhalation allergy than in those without (aOR = 5.65; 95%CI 1.43–22.22; *p* = 0.01).

On the other hand, longer duration of breastfeeding reduced the risk of allergy and sensitization (aOR = 0.82; 95%CI 0.69–0.99; *p* = 0.008). Earlier introduction of a varied diet significantly reduced the risk of allergies. The introduction of more product groups into the child’s diet at 6 months of age (5–6 vs. 0–4 or >7 vs. 0–4) reduced the risk of allergy; aOR = 0.16 (95%CI 0.04–0.67) (*p* = 0.003) and aOR = 0.15 (95%CI 0.01–1.67) (*p* = 0.001), respectively (Table 5).

Regarding the other confounders, the logistic regression found a number of other factors to not be significantly associated with the risk of allergy and/or sensitization: the course of pregnancy, duration of pregnancy, mode of delivery, birth weight, having siblings, sex, number of children in the family, place of living, order of birth, parental education, parent age.

## 4. Discussion

Following the increasing trend of allergic diseases, a number of studies have examined their development, with many focusing on the influence of diet. Among these, this is the first study to analyze the importance of diet in the development of allergies in a group of Polish children.

Our findings do not indicate that the type of complementary foods (including allergenic ones) or the time of their introduction influence the development of allergies and sensitization in children under three years of age. In all studied groups of children, complementary products were most often introduced at 7 to 12 months of age.

Literature data on the impact of exposure time to food allergens on the development of allergies are varied. Similar to our study, the GUSTO study (Growing Up in Singapore Towards healthy Outcomes) found no relationship between the time of introducing complementary foods and the occurrence of allergies [16]. The study followed mother–infant pairs from pregnancy through early childhood, and the analysis included 1152 women of Chinese, Malay, and Indian descent [16]. More specifically, the findings did not show any statistically significant relationship between the delayed introduction of eggs, peanuts, and shellfish and the development of allergies in children observed up to 24 months of age, even in children from the risk group.

Another study analyzing various methods of food allergy prophylaxis found that, apart from breastfeeding, the only effective intervention in children is the induction of oral tolerance [17]. One of the most significant food allergens in children is milk. Katz et al. propose that early exposure to milk protects against the development of cow’s milk protein allergy (CMPA). Therefore, in order to promote oral tolerance, especially in the group of children with an increased risk of allergies, complementary feeding with milk before four months of age should be considered [11]. However, although observational studies indicate that early introduction of cow’s milk may protect against the development of CMPA, the results of randomized controlled trials are conflicting; indeed, some expert recommendations and recent findings indicate that exposure to cow’s milk in the first week of life, especially the first two days, may increase the risk of CMPA [12,13]. Interestingly, it was also found that regular use of milk formulas, even in small amounts, may give better results in an infant previously exposed to cow’s milk proteins in the first days of life, than a complete lack of exposure [18]. However, it must be emphasized that incidental supplementation with formula milk in the first three days of life may result in a significant increase in the risk of developing CMPA in early childhood [17].

Based on the consensus regarding primary prophylaxis of food allergy by Fleischer et al., it is recommended that hen’s eggs should be introduced to all infants, regardless of the risk of developing allergies, in the first six months, but not earlier than four months. It is also recommended that the eggs should be boiled or baked, but not served raw; in addition, after the eggs are introduced into the child’s diet, it is important to provide a regular supply [12].

Similar recommendations apply to the introduction of peanuts. It is advisable to include peanuts in the diet of all children, regardless of the risk of developing allergies, in the first six months, but not earlier than four months. Again, it is important to continue the supply of peanuts after first exposure [12]. So far, numerous recommendations have been published regarding the introduction of peanuts into the diet with the aim of preventing peanut allergy [12,13,19].

Interestingly, our present findings contradict those of previous studies indicating that the introduction of peanuts and eggs into the diet at a certain time prevents the occurrence of allergies. However, the observed lack of significant differences in our present findings may be due to the small size of our patient group.

In addition, unlike peanuts and hen’s eggs, no reliable evidence exists regarding the specific recommendations concerning the introduction of other allergenic complementary foods to the infant’s diet, such as milk, soya, wheat, tree nuts, sesame, fish, or crustaceans. It is known, however, that delaying their introduction may not bring the expected results and can even be harmful [12].

Both the timing of the introduction of food products and their frequency of consumption can affect the occurrence of allergies and/or sensitization. However, it has been emphasized that, once a product has been introduced, it should be regularly consumed [12].

In our study, a significant difference was found only in the frequency of fish consumption. Children with allergic symptoms were given fish significantly more often, usually 1–3x/week, than children without allergic symptoms, who received fish at most 1–3x/month. Our present findings contradict those of previous studies indicating that fish consumption appears to have a protective effect against the development of allergies; this has been attributed to the anti-inflammatory properties of omega-3 fatty acids, naturally occurring in large amounts in fish [20].

The different results of our research compared to others can be explained by the reverse causation, i.e., children with allergic symptoms more often received allergenic foods, perhaps for prophylactic purposes. Ultimately, however, it did not protect these children from the occurrence of allergies, because perhaps such a procedure was implemented too late; or maybe other factors, including genetic ones, could have played a role.

Based on the frequency of food consumption questionnaire, Maslin et al. showed no difference in the frequency of the consumption of fruit, vegetables, fish, or cereals between children with CMPA and those without [21]. Andrusaityte et al. [22] examined the relationship between the frequency of food consumption and the development of allergies. They assessed the relationship between the frequency of consumption of fruits, vegetables, nuts, meat, and fish and the incidence of wheezing, asthma and AD among 1489 preschool children living in Lithuania. It was shown that 83.3% of all children consumed fresh fruit and/or vegetables at least three times a week. Fruit consumption was found to be associated with a reduced risk of wheezing (aOR = 0.48; 95%CI 0.22–0.96). Similarly, more frequent consumption of nuts was also associated with a 61% lower risk of AD in the study children. The authors indicate that frequent consumption of fresh fruit and nuts appears to have a beneficial effect on the occurrence of allergies among children. However, the frequency of fish consumption appeared to have no significant effect [22].

The frequency of eating food allergens may be related to the development of allergies. Although this was not demonstrated in our study, it is known that regular exposure to a given food has a preventive effect. It seems that the allergen dose, which will increase with greater consumption, plays an important role in inducing food tolerance, although its precise value has not yet been established.

In addition, the development of allergies and/or sensitization may be influenced by the method of food processing; this may change the allergenicity of the food due to the modification of the physicochemical properties of its proteins. Such modifications can be further influenced by factors such as processing conditions, type of food, and allergen content [23].

In our study, heat-treated peanuts were consumed significantly more often in children without allergy and sensitization than those with sensitization and allergy symptoms; the latter group were more likely to eat the raw form (*p* = 0.02). The degree of processing of peanuts is relevant to the occurrence of peanut allergy. Peanut proteins demonstrate increasing allergenicity under the influence of temperature. During heat treatment and digestion, peanut allergens undergo certain modifications, such as acylation, polymerization, nitration, and the Maillard reaction, which affects their potential to cause allergy symptoms. For example, roasting peanuts at high temperatures probably promotes the formation of compact globular protein aggregates, which increases their allergenicity [24]. While peanuts are mainly eaten roasted, as peanut butter or peanut crisps in Europe and the US, they are more often used in cooked dishes in Asia [25]. Unlike other food allergens, such as milk or egg, cooked peanuts are less allergenic than roasted peanuts. It has been shown that cooking reduces the binding capacity of specific IgE antibodies to peanut proteins in vitro: boiling peanuts for 20 min in water (100 °C) reduced the binding capacity of sIgE to all tested peanut allergens (Ara h 1, 2 and 3) [26]. In addition, Tao et al. report that intensive cooking for up to 12 h reduces the ability of peanuts to bind to IgE, while maintaining their ability to stimulate cells [27]. Peanut oil, produced from peanuts in unrefined form, contains a relatively large amount of protein, which may induce allergic reactions; however, the refined oil does not cause any reaction, even in people who have experienced symptoms of anaphylaxis after eating a peanut [28,29]. Although these findings are not fully confirmed by our present data, this could result from the influence of other factors on the development of allergies than the degree of processing of peanuts per se.

Our study has some weaknesses, such as the small size of the study group and the fact that only three time intervals were analyzed, i.e., foods introduced at ≤3 months, 4–6 months, and 7–12 months. Additionally, the limitation of the study is a lack of analysis of the first days of life. The questionnaire was very extensive and it was necessary to limit the number of questions. However, despite this, the study has a number of strengths, not least that it serves as a prospective analysis of the relationship between diet and allergy in a population of children in Poland. Even so, to obtain more reliable data regarding the relationship between the time of introducing complementary foods in children and the development of allergies and sensitization, further studies are needed based on larger groups of participants.

## 5. Conclusions

Neither the time of introduction of complementary foods, including allergenic forms, or their type appear to have any effect on the development of allergies and sensitization in children under three years of age. The degree of processing and the frequency of supply of products may affect the development of allergies and sensitization. However, there is a need for further research on the relationship between diet and the possibility of preventing food allergy in children.

## Figures and Tables

**Figure 1 nutrients-15-02054-f001:**
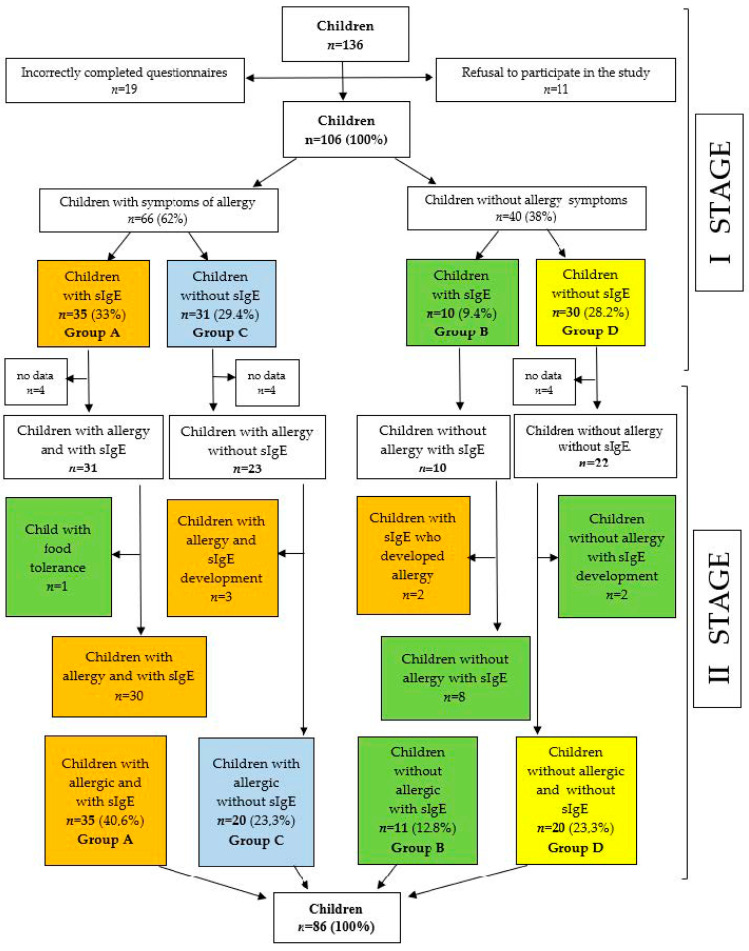
Flow chart of the study design.

**Figure 2 nutrients-15-02054-f002:**
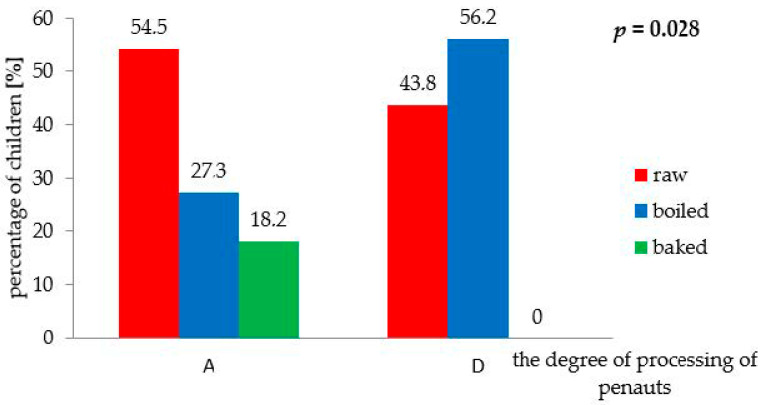
The degree of processing of peanuts administered in the children with allergic symptoms and sensitization (group A) and in the children without allergic symptoms or sensitization (group D).

**Figure 3 nutrients-15-02054-f003:**
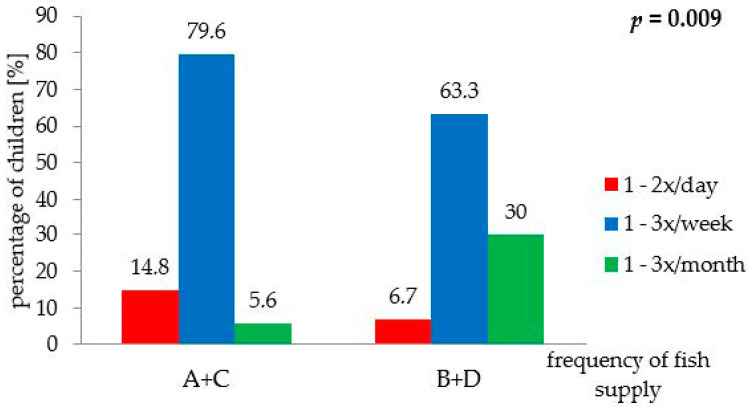
The frequency of fish supply in the group with symptoms of allergy (Groups A + C) and without allergic symptoms (Groups B + D).

**Table 1 nutrients-15-02054-t001:** Characteristics of the study and control groups.

Parameter		Total *n* = 86 (100%)	Group A *n* = 35 (40.6%)	Group B *n* = 11 (12.8%)	Group C *n* = 20 (23.3%)	Group D *n* = 20 (23.3%)	*p*
The course of pregnancy, *n* (%)							
	normal	80 (93.1)	33 (94.2)	11 (100)	19 (95)	17 (85)	>0.05
	endangered pregnancy	6 (6.9)	2 (5.7)	0 (0)	1 (5)	3 (15)	
Type of delivery, *n* (%)							>0.05
	vaginal	55 (63.9)	23 (65.7)	7 (63.6)	11 (55)	14 (70)	
	caesarian section	31 (36.1)	12 (34.3)	4 (36.4)	9 (45)	6 (30)	
Duration of pregnancy, Hbd							>0.05
median (Q1–Q3)		39 (38–40)	39 (38–41)	40 (39.5–40)	38.5 (36–40)	39 (38–40)	
Birth weight (grams)							>0.05
mean ± SD		3256 ± 591.9	3376 ± 586.2	3330 ± 495.5	3096 ± 742.9	3177 ± 550.5	
Points on the Apgar scale							>0.05
median (Q1–Q3)		10 (10–10)	10 (10–10)	10 (10–10)	10 (8.5–10)	10 (10–10)	
Age of the child, month							>0.05
mean ± SD		31.9 ± 3.5	31.9 ± 3.8	32.5 ± 3.2	31.9 ± 3.2	31.9 ± 3.2	
Sex, *n* (%)							>0.05
	boys	50 (58.1)	23 (65.7)	5 (45.5)	8 (40)	14 (70)	
	girls	36 (41.9)	12 (34.3)	6 (54.5)	12 (60)	6 (30)	
Having siblings, *n* (%)							>0.05
	yes	39 (45.3)	18 (51.4)	6 (54.5)	14 (70)	9 (45)	
	no	47 (54.7)	17 (48.6)	5 (45.5)	6 (30)	11 (55)	
Number of children in the family, *n* (%)							>0.05
	1	47(54.6)	18 (51.4)	6 (54.5)	14 (70)	9 (45)	
	2	28 (32.6)	13 (37.1)	2 (18.2)	4 (20)	9 (45)	
	3	7 (8.1)	3 (8.6)	2 (18.2)	2 (10)	0 (0)	
	4	4 (4.7)	1 (2.9)	1 (9.1)	0 (0)	2 (10)	
Order of birth, *n* (%)							>0.05
	first	49 (57.0)	19 (54.3)	7 (63.6)	14 (70)	9 (45)	
	the second	26 (30.2)	12 (34.3)	1 (9.1)	4 (20)	9 (45)	
	third	7 (8.1)	3 (8.6)	2 (18.2)	2 (10)	0 (0)	
	fourth	4 (4.7)	1 (2.9)	1 (9.1)	0 (0)	2 (10)	
Place of living, *n* (%)							0.05
	town of ≥100 thousand inhabitants	24 (43.6)	13 (37.1)	2 (18.2)	5 (25)	11 (55)	
	town of <100 thousand inhabitants	14 (25.4)	12 (34.3)	2 (18.2)	6 (30)	2 (10)	
	village	17 (31.0)	10 (28.6)	7 (62.6)	9 (45)	7 (35)	
Parent age, years							>0.05
	mother mean ± SD	30.3 ± 4.9	30.7 ± 4.6	29.8 ± 5.5	33.2 ± 5.2	29.6 ± 5.6	
	father mean± SD	32.7 ± 4.6	32.8 ± 4.6	32.8 ± 6.2	35.9 ± 6.1	32.5 ± 4.6	
Parent’s education, *n* (%)							>0.05
mother							
	basic	5 (9.1)	0 (0)	3 (27.3)	1 (5)	5 (25)	
	secondary	12 (21.8)	10 (28.6)	2 (18.2)	1 (5)	2 (10)	
	higher	38 (69.1)	25 (71.4)	6 (54.5)	18 (90)	13 (65)	
father							>0.05
	basic	10 (18.2)	6 (17.1)	4 (36.4)	2 (10)	4 (20)	
	secondary	14 (25.4)	9 (25.7)	3 (27.3)	5 (25)	5 (25)	
	higher	31 (5.4)	20 (57.1)	4 (36.4)	13 (65)	11 (55)	
Family history of atopy		59 (68.6)	28 (80)	5 (45.5)	15 (75)	11 (55)	0.0002
Allergies in family members							>0.05
	mother	18 (20.9)	10 (28.6)	1 (9.1)	3 (15)	4 (20)	
	father	26 (30.2)	13 (37.1)	3 (27.3)	5 (25)	5 (25)	
	both parents	17 (19.8)	8 (22.9)	0 (0)	7 (35)	2 (10)	
	siblings	12 (13.9)	5 (14.3)	3 (23.3)	2 (10)	2 (10)	
Type of allergy in the family							>0.05
	food allergy	29 (33.7)	13 (37.1)	4 (36.4)	7 (35)	5 (25)	
	atopic dermatitis	23 (26.7)	13 (37.1)	0 (0)	7 (35)	3 (15)	
	inhalant allergy	49 (57)	27 (77.1)	4 (36.4)	10 (50)	8 (40)	

**Table 2 nutrients-15-02054-t002:** Time of introducing complementary foods in studied groups of children.

Time of Introducing Complementary Foods	Group A + C *n* = 55 (100%)	Group B + D *n* = 31 (100%)	*p*	Group A *n* = 35 (100%)	Group D *n* = 20 (100%)	*p*	Group A + B *n* = 46 (100%)	Group C + D *n* = 40 (100%)	*p*
Egg white			0.89			1.00			0.53
≤3 m	3 (5.5)	2 (6.5)		3 (8.6)	1 (5)		4 (8.7)	1 (2.5)	
4–6 m	5 (9.1)	2 (6.5)		3 (8.6)	1 (5)		4 (8.7)	3 (7.5)	
7–12 m	47 (85.5)	27 (87.1)		29 (82.9)	18 (90)		38 (82.6)	36 (90)	
Milk			0.64			0.82			0.63
≤3 m	4 (7.4)	1 (3.3)		3 (8.8)	1 (5.3)		3 (6.7)	2 (5.1)	
4–6 m	15 (27.8)	11 (36.7)		9 (26.5)	4 (21.1)		16 (35.6)	10 (25.6)	
7–12 m	35 (64.8)	18 (60)		22 (64.7)	14 (73.7)		26 (57.8)	27 (69.2)	
Peanuts			0.66			1.00			1.00
≤3 m	0 (0)	0 (0)		0 (0)	0 (0)		0 (0)	0 (0)	
4–6 m	2 (13.3)	4 (23.5)		2 (18.2)	4 (25)		2 (16.7)	4 (20)	
7–12 m	13 (86.7)	13(76.5)		9 (81.8)	12 (75)		10 (83.3)	16 (80)	
Wheat			0.59			0.09			0.06
≤3 m	2 (3.7)	1 (3.3)		2 (5.7)	1 (5.3)		2 (4.3)	1 (2.6)	
4–6 m	11 (20.4)	10 (33.3)		4 (11.4)	7 (36.8)		7 (15.2)	14 (36.8)	
7–12 m	41 (75.9)	19 (63.3)		29 (82.9)	11 (57.9)		37 (80.4)	23 (60.5)	
Soybean			1.00			0.56			0.26
≤3 m	0 (0)	0 (0)		0 (0)	0 (0)		0 (0)	0 (0)	
4–6 m	2 (28.6)	4 (33.3)		2 (50)	3 (27.3)		3 (60)	3 (21.4)	
7–12 m	5 (71.4)	8 (66.7)		2 (50)	8 (72.7)		2 (40)	11 (78.6)	
Fish			0.42			0.43			0.21
≤3 m	5 (9.3)	1 (3.3)		2 (5.9)	1 (5.3)		2 (4.4)	4 (10.3)	
4–6 m	11 (20.4)	9 (30)		5 (14.7)	6 (31.6)		8 (17.8)	12 (30.8)	
7–12 m	38 (70.4)	20(66.7)		27 (79.4)	12 (63.2)		35 (77.8)	23 (59)	
Tree nuts			1.00			1.00			0.62
≤3 m	0 (0)	0 (0)		0 (0)	0 (0)		0 (0)	0 (0)	
4–6 m	2 (22.2)	2 (16.7)		2 (25)	2 (16.7)		2 (25)	2 (15.4)	
7–12 m	7 (77.8)	10 (83.3)		6 (75)	10 (83.3)		6 (75)	11 (84.6)	
Shellfish			>0.05			>0.05			>0.05
≤3 m	0 (0)	0 (0)		0 (0)	0 (0)		0 (0)	0 (0)	
4–6 m	0 (0)	0 (0)		0 (0)	0 (0)		0 (0)	0 (0)	
7–12 m	2 (100)	4 (100)		2 (100)	4 (100)		2 (100)	4 (100)	

m—month of life; Groups A + C—Children with allergy symptoms; Groups B = D—Children without allergic symptoms; Group A—Children with allergic symptoms and concomitant sensitization; group D—Children without allergic symptoms and without concomitant sensitization; Groups A + B—Children with sensitization; Groups C + D—Children without sensitization.

**Table 3 nutrients-15-02054-t003:** The frequency and form of supply of allergenic foods.

Frequency and Form of Food Supply	Groups A + C *n* = 55 (100%)	Groups B + D *n* = 31 (100%)	*p*	Group A *n* = 35 (100%)	Group D *n* = 20 (100%)	*p*	Groups A + B *n* = 46 (100%)	Groups C + D *n* = 40 (100%)	*p*
Frequency of egg white supply, *n* (%)			0.083			0.15			0.845
1–2x/day	20 (37)	5 (16.1)		11 (32.4)	2 (10)		14 (31.1)	11 (27.5)	
1–3x/week	27 (50)	23 (74.2)		18 (52.9)	16 (80)		25 (55.6)	25 (62.5)	
1–3x/month	7 (13)	3 (9.7)		5 (14.7)	2 (10)		6 (13.3)	4 (10)	
Form of egg protein supply, *n* (%)			0.066			0.189			0.484
raw	1 (1.8)	3 (9.7)		1 (2.9)	2 (10)		2 (4.3)	2 (5)	
boiled	49 (89.1)	28 (90.3)		30 (85.7)	18 (90)		40 (87)	37 (92.5)	
baked	5 (9.1)	0 (0)		4 (11.4)	0 (0)		4 (8.7)	1 (2.5)	
Frequency of milk supply, *n* (%)			0.715			0.248			0.269
1–2x/day	37 (68.5)	18 (60)		23 (67.6)	9 (47.4)		32 (71.1)	23 (59)	
1–3x/week	13 (24.1)	10 (33.3)		7 (20.6)	8 (42.1)		9 (20)	14 (35.9)	
1–3x/month	4 (7.4)	2 (6.7)		4 (11.8)	2 (105)		4 (8.9)	2 (5.1)	
Form of milk supply, *n* (%)			0.024			0.15			1.00
raw	9 (16.7)	0 (0)		5 (14.7)	0 (0)		5 (11.1)	4 (10.5)	
boiled	45 (83.3)	29 (100)		29 (85.3)	18 (100)		40 (88.9)	34 (89.5)	
baked	0 (0)	0 (0)		0 (0)	0 (0)		0 (0)	0 (0)	
Frequency of peanuts supply, *n* (%)			>0.05			>0.05			>0.05
1–2x/day	1 (6.7)	0 (0)		1 (9.1)	0 (0)		1 (8.3)	0 (0)	
1–3x/week	2 (13.3)	1 (5.9)		2 (18.2)	0 (0)		3 (25)	0 (0)	
1–3x/month	12 (80)	16 (94.1)		8 (72.7)	16 (100)		8 (66.7)	20 (100)	
Form of peanuts supply, *n* (%)			0.07			0.028			0.108
raw	7 (46.7)	7 (41.2)		6 (54.5)	7 (43.8)		6 (50)	8 (40)	
boiled	5 (33.3)	9 (52.9)		3 (27.3)	9 (56.2)		4 (33.3)	10 (50)	
baked	3 (20)	1 (5.9)		2 (18.2)	0 (0)		2 (16.7)	2 (10)	
Frequency of wheat supply, *n* (%)			>0.05			0.267			>0.05
1–2x/day	32 (60.4)	15 (50)		20 (58.8)	8 (42.1)		27 (60)	20 (52.6)	
1–3x/week	20 (37.7)	14 (46.7)		14 (41.2)	11 (57.9)		17 (37.8)	17 (44.7)	
1–3x/month	1 (1.9)	1 (3.3)		0 (0)	0 (0)		1 (2.2)	1 (2.6)	
Form of wheat supply, *n* (%)			>0.05			>0.05			>0.05
boiled	26 (49)	7 (23.3)		15 (44.1)	3 (15.8)		19 (42.2)	15 (39.5)	
baked	27 (51)	23 (76.7)		19 (55.9)	16 (84.2)		26 (57.8)	23 (60.5)	
Frequency of soy supply, *n* (%)			>0.05			>0.05			>0.05
1–2x/day	0 (0)	2 (16.7)		0 (0)	2 (18.2)		0 (0)	2 (14.3)	
1–3x/week	1 (14.3)	7 (58.3)		0 (0)	7 (63.6)		0 (0)	8 (57.1)	
1–3x/month	6 (85.7)	3 (25)		4 (100)	2 (18.2)		5 (100)	4 (28.6)	
Form of soy supply, *n* (%)			0.074			0.56			1.00
boiled	5 (71.4)	3 (25)		2 (50)	3 (27.3)		2 (40)	6 (42.9)	
baked	2 (28.6)	9 (75)		2 (50)	8 (72.7)		3 (60)	8 (57.1)	
Frequency of fish supply, *n* (%)			0.009			0.251			0.943
1–2x/day	8 (14.8)	2 (6.7)		4 (11.8)	1 (5.3)		5 (11.1)	5 (12.8)	
1–3x/week	43 (79.6)	19 (63.3)		28 (82.4)	14 (73.7)		33 (73.3)	29 (74.4)	
1–3x/month	3 (5.6)	9 (30)		2 (5.9)	4 (21.1)		7 (15.6)	5 (12.8)	
Form of fish supply, *n* (%)			0.155			0.496			0.787
boiled	46 (85.2)	21 (70)		28 (82.4)	14 (73.7)		35 (77.8)	32 (82.1)	
baked	8 (14.8)	9 (30)		6 (17.6)	5 (26.3)		10 (22.2)	7 (17.9)	
Frequency of tree nuts supply *n* (%)			>0.05			>0.05			>0.05
1–2x/day	2 (22.2)	0 (0)		2 (25)	0 (0)		2 (25)	0 (0)	
1–3x/week	2 (22.2)	6 (50)		2 (25)	6 (50)		2 (25)	6 (46.2)	
1–3x/month	5 (55.6)	6 (50)		4 (50)	6 (50)		4 (50)	7 (53.8)	
Form of tree nuts supply *n* (%)			>0.05			>0.05			>0.05
raw	7 (77.8)	8 (66.7)		6 (75)	8 (66.7)		6 (75)	9 (69.2)	
boiled	2 (22.2)	0 (0)		2 (25)	0 (0)		2 (25)	0 (0)	
baked	0 (0)	4 (33.3)		0 (0)	4 (33.3)		0 (0)	4 (30.8)	
Frequency of shellfish supply *n* (%)			>0.05			>0.05			>0.05
1–2x/day	0 (0)	0 (0)		0 (0)	0 (0)		0 (0)	0 (0)	
1–3x/week	0 (0)	0 (0)		0 (0)	0 (0)		0 (0)	0 (0)	
1–3x/month	2 (100)	4 (100)		2 (100)	4 (100)		2 (100)	4 (100)	
Form of shellfish supply *n* (%)			1.00			1.00			1.00
boiled	2 (100)	3 (75)		2 (100)	3 (75)		2 (100)	3 (75)	
baked	0 (0)	1 (25)		0 (0)	1 (25)		0 (0)	1 (25)	

**Table 4 nutrients-15-02054-t004:** Logistic regression analysis of factors related to the occurrence of allergies in children with allergic symptoms (Groups A + C) vs. children without allergic symptoms (Groups B + D), regardless of sensitization.

Parameter	Model I		Model II	
	OR (95%CI)	*p*	aOR (95%CI)	*p*
Family history of inhalant allergy yes vs. no	3.25 (1.3–8.14)	0.012		
The occurrence of AD in the family yes vs. no	5.3 (1.4–19.8)	0.012	17.6 (1.9–159.6)	0.011
The frequency of egg white supply 1–3x/week vs. 1–2x/day	0.29 (0.1–0.91)	0.033		
Frequency of fish supply 1–3x/month vs. 1–2x/day	0.08 (0.01–0.63)	0.016		
Frequency of supply of gluten-free products 1–3x/week vs. 1–2x/day	0.34 (0.13–0.9)	0.03		
Using ready-made products yes vs. no	9.49 (3.27–27.59)	<0.001	11.5 (2.7–49.7)	0.001
Variety of diet (number of product groups)				
6 months 5–6 vs. 0–4	0.22 (0.08–0.62)	0.004	0.17 (0.04–0.71)	0.015
6 months ≥7 vs. 0–4	0.1 (0.02–0.45)	0.003	0.08 (0.01–0.58)	0.012
12 months ≥8 vs. 5–7	0.18 (0.07–0.48)	0.001	0.14 (0.03–0.57)	0.006

**Table 5 nutrients-15-02054-t005:** Logistic regression analysis of factors related to the occurrence of allergies and sensitization in children with allergy symptoms and coexisting sensitization (Group A) vs. children without allergy and without coexisting sensitization (Group D).

Parameter	Model I		Model II	
	OR (95%CI)	*p*	aOR (95%CI)	*p*
Family history of food allergy yes vs. no	3.51 (1.1–11.15)	0.034		
Family history of inhalant allergy yes vs. no	6.32 (2.13–18.73)	0.001	5.65 (1.43–22.22)	0.001
The occurrence of AD in the family yes vs. no	5.3 (1.4–20.03)	0.014		
Exclusive breastfeeding time	0.81 (0.7–0.95)	0.009	0.82 (0.69–0.99)	0.008
Time to introduce foods in the form of lumps 7.5 months vs. 8 months	1.49 (1.02–2.16)	0.038		
Time to introduce solid foods 10 months vs. 11 months	1.33 (1.04–1.71	0.025		
Variety of diet (number of product groups)				
6 months 5–6 vs. 0–4	0.16 (0.05–0.54)	0.003		
6 months ≥7 vs. 0–4	0.06 (0.01–0.42)	0.005	0.16 (0.04–0.67)	0.003
12 months ≥8 vs. 5–7	0.11 (0.03–0.34)	<0.001	0.15 (0.01–1.67)	0.001

## Data Availability

Not applicable.

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
