# Peer review of "Introduction of Complementary Foods and the Risk of Sensitization and Allergy in Children up to Three Years of Age"

_nutrients, 2023, doi:10.3390/nu15092054_

Round 1
Reviewer 1 Report
The authors present a complete and very interesting statistical analysis about introduction of complementary foods and the risk of 2 sensitization and allergy in children up to three years of age. The study showed concordant results with other data in the literature. The data is treated appropriately.
Authors are requested to replace Figure 1 with a sharper image because the uploaded image is blurry.
Author Response
Answer to reviewer 1.
April 19, 2023
Nutrients
To Whom This May Concern:
We attach our revision of the manuscript (ID: nutrients-2340701) entitled "Introduction of complementary foods and the risk of sensitization and allergy in children up to three years of age”.
We really do appreciate Yours careful reading of our manuscript. We acknowledge the Reviewer’s comments and suggestions very much, which are valuable in improving the quality of our manuscript.
Here are our replies to the comments:
- Authors are requested to replace Figure 1 with a sharper image because the uploaded image is blurry.
Figure 1 has been improved, the image quality is better.
I would like once again to thank you for all your help and support. I truly believe your comments have helped make this article more valuable. I hope that this manuscript will be now accepted for publication in Nutrients, which would be considered a great honor for me. Thank You for Your time and consideration.
With best regards,
Magdalena Chęsy
Aneta Krogulska

Reviewer 2 Report
Thank you for the article.
The article is looking at the timing or introduction of complementary food and the type and frequency of the food on the development of food allergy.
For the methodology, more details should be given. When were the Food questionnaires administered? When was the sIgE panel done on these children? At what age? Was it just done once? For those who were found to be sensitized, were these children challenged to confirm allergy? Any reason why shellfish and tree nuts were not included into the food panel? When were the characterization into the 4 groups done? only at 30 months based on clinical symptoms? Why did you not look at the subjects at stage 1 as well?
In terms of the results, you found that there was no difference between timing of introduction of food and the type and frequency on food allergy development. It was not clear in the article, what form of egg white was given. Was it well cooked egg or baked egg and what quantities? For milk, it will be good to know whether any of these children had received formula for the first few days of life before being subsequently exclusively breastfed. It is not clear how they were fed but one is assuming that they were exclusively breastfed so you may want to clarify this as well. Also whether any of these children received intermittent supplementation of formula in the first few months of life. Many of these children are firstborn infants, so it may take time for mothers to establish breastfeeding. This is more so as you have made some comments about this in the discussion. In terms of fish, you mentioned that more frequent feeding of fish was associated with development of fish allergy which is contrary to current understanding that regular feeding of the food will lead to tolerance. When you look at the graph presented, the majority of children who had fish allergy as well as those who did not, had regular intake of fish at 1-3x/week. Were these children allergic to all types of fish or only to specific fish? For peanut, what forms of raw peanuts do parents in Poland give? Most of the cohort studies (real life), the peanut is cooked/roasted.
In terms of food diversity, you mentioned that early introduction of a varied diet reduced the risk of food allergy. This was probably true at 6 months and 12 months but not at 3 months. Would you like to explain why this might be?
In terms of risk factors for food allergy, you have identified family history of inhalation allergy. Did you look at the children who had eczema and their risk of food allergy and especially whether the severity of the eczema was associated with a higher risk?
Author Response
Answer to reviewer 2.
April 19, 2023
Nutrients
To Whom This May Concern:
We attach our revision of the manuscript (ID: nutrients-2340701) entitled "Introduction of complementary foods and the risk of sensitization and allergy in children up to three years of age”.
We really do appreciate Yours careful reading of our manuscript. We acknowledge the Reviewer’s comments and suggestions very much, which are valuable in improving the quality of our manuscript.
Here are our replies to the comments:
- When were the Food questionnaires administered?
The questionnaire has been forwarded to the children's legal guardians after they were qualified for the study. The children were then 6-18 months old (some mothers completed the questionnaire retrospectively, and some prospectively; it seems that the period of the child's life analyzed was so close (in the case of mothers completing the questionnaire retrospectively) that most mothers remembered their diet in 1 year).
- When was the sIgE panel done on these children? At what age? Was it just done once?
The study was performed in 2 stages. In the first stage, the study included children aged 6 to 18 months, in the second stage, the same children were examined after another 12 months.
In order to assess asIgE, blood was collected at 2 time points: in the first stage in children aged 12-18 months, and in the second stage after another 12-18 months, i.e. in children aged 24-36 months.
Data from the second stage were analyzed (explanation below).
- For those who were found to be sensitized, were these children challenged to confirm allergy?
Children with a clear history (i.e. children with anaphylaxis or anaphylactic reaction after ingestion of a specific allergen) and confirmed sensitization to a given food allergen were not challenged. Oral provocation tests were performed on allergic children with no clear medical history. In contrast, children with non-IgE-independent milk or egg allergy were put on a diagnostic elimination diet and then re-exposed to the allergen in the diet. In the case of improvement on the elimination diet, and then deterioration after the inclusion of the allergen in the diet, food allergy was confirmed.
- Any reason why shellfish and tree nuts were not included into the food panel?
Due to financial constraints, only 10 food allergens and 10 inhalant allergens were selected for the sIgE assessment. The assessment of sIgE for shellfish and tree nuts was abandoned due to the fact that allergy to these products in small children is extremely rare in Poland.
- When were the characterization into the 4 groups done? only at 30 months based on clinical symptoms?
Initially, patients were divided into two groups based on clinical symptoms: children with allergic symptoms (n=66; 62%) and without (n=40; 38%). After obtaining the sIgE results, children with and without sensitization were distinguished.
Finally, based on the assessment by an allergist and the results of sIgE, the examined children were divided into 4 subgroups. The division into groups was made at two time points in the first stage of the study at the age of 12-18 months, and in the second stage after another 12-18 months, i.e. in children aged 24-36 months.
- Why did you not look at the subjects at stage 1 as well?
Due to the limited number of words, data on the first stage of the study are not included in the presented manuscript. The conducted study was the subject of the doctoral dissertation (the first and second stages of the study were discussed), which consisted of 305 pages, 94 tables and 63 figures.
- It was not clear in the article, what form of egg white was given. Was it well cooked egg or baked egg and what quantities?
In all the studied groups, parents most often served boiled egg whites to their children. In the conducted study, no question was asked about the amount of egg white administered, but the frequency of supply of this product was taken into account. In all study groups, egg white was usually administered 1-3x/week. Detailed data are presented in the table below. We added Table 3 into the manuscript.
Frequency and form of food supply |
Group A+C n=55 (100%) |
Group B+D n=31 (100%) |
P |
Group A n=35 (100%) |
Group D n=20 (100%) |
P |
Group A+B |
Group C+D n=40 (100%) |
P |
Frequency of egg white supply n (%) 1-2x/day 1-3x/week 1-3x/month |
20 (37) 27 (50) 7 (13) |
5 (16.1) 23 (74.2) 3 (9.7) |
0.083 |
11 (32.4) 18 (52.9) 5 (14.7) |
2 (10) 16 (80) 2 (10) |
0.15 |
14 (31.1) 25 (55.6) 6 (13.3) |
11 (27.5) 25 (62.5) 4 (10) |
0.845 |
Form of egg protein supply n (%) raw boiled baked |
1 (1.8) 49 (89.1) 5 (9.1) |
3 (9.7) 28 (90.3) 0 (0) |
0.066 |
1 (2.9) 30 (85.7) 4 (11.4) |
2 (10) 18 (90) 0 (0) |
0.189 |
2 (4.3) 40 (87) 4 (8.7) |
2 (5) 37 (92.5) 1 (2.5) |
0.484 |
Frequency of milk supply n (%) 1-2x/day 1-3x/week 1-3x/month |
37 (68.5) 13 ( 24.1 ) 4 (7.4) |
18 (60) 10 (33.3) 2 (6.7) |
0.715 |
23 (67.6) 7 (20.6) 4 (11.8) |
9 (47.4) 8 (42.1) 2 (105) |
0.248 |
32 (71.1) 9 ( 20 ) 4 ( 8,9 ) |
23 (59) 14 (35.9) 2 (5.1) |
0.269
|
Form of milk supply n (%) raw boiled baked |
9 (16.7) 45 (83.3) 0 (0) |
0 (0) 29 (100) 0 (0) |
0.024
|
5 (14.7) 29 (85.3) 0 (0) |
0 (0) 18 (100) 0 (0) |
0.15 |
5 (11.1) 40 (88.9) 0 (0) |
4 (10.5) 34 (89.5) 0 (0) |
1.00 |
Frequency of peanuts supply n (%) 1-2x/day 1-3x/week 1-3x/month |
1 (6.7) 2 (13.3) 12 (80) |
0 (0) 1 (5.9) 16 (94.1) |
>0.05 |
1 (9.1) 2 (18.2) 8 (72.7) |
0 (0) 0 (0) 16 (100) |
>0.05 |
1 (8.3) 3 (25) 8 (66.7) |
0 (0) 0 (0) 20 (100) |
>0.05
|
Form of peanuts supply n (%) raw boiled baked |
7 (46.7) 5 (33.3) 3 (20) |
7 (41.2) 9 (52.9) 1 (5.9) |
0.07 |
6 (54.5) 3 (27.3) 2 (18.2) |
7 (43.8) 9 (56.2) 0 (0) |
0.028 |
6 (50) 4 (33.3) 2 (16.7) |
8 (40) 10 (50) 2 (10) |
0.108 |
Frequency of wheat supply n (%) 1-2x/day 1-3x/week 1-3x/month |
32 (60.4) 20 (37.7) 1 (1.9) |
15 (50) 14 (46.7) 1 (3.3) |
>0.05 |
20 (58.8) 14 (41.2) 0 (0) |
8 (42.1) 11 (57.9) 0 (0) |
0.267 |
27 (60) 17 (37.8) 1 (2.2) |
20 (52.6) 17 (44.7) 1 (2.6) |
>0.05 |
Form of wheat supply n (%) boiled baked |
26 (49) 27 (51) |
7 (23.3) 23 (76.7) |
>0.05 |
15 (44.1) 19 (55.9) |
3 (15.8) 16 (84.2) |
>0.05 |
19 (42.2) 26 (57.8) |
15 (39.5) 23 (60.5) |
>0.05 |
Frequency of soy supply n (%) 1-2x/day 1-3x/week 1-3x/month |
0 (0) 1 (14.3) 6 (85.7) |
2 (16.7) 7 (58.3) 3 (25) |
>0.05 |
0 (0) 0 (0) 4 (100) |
2 (18.2) 7 (63.6) 2 (18.2) |
>0.05 |
0 (0) 0 (0) 5 (100) |
2 (14.3) 8 (57.1) 4 (28.6) |
>0.05 |
Form of soy supply n (%) boiled baked |
5 (71.4) 2 (28.6) |
3 (25) 9 (75) |
0.074 |
2 (50) 2 (50) |
3 (27.3) 8 (72.7) |
0.56 |
2 (40) 3 (60) |
6 (42.9) 8 (57.1) |
1.00 |
Frequency of fish supply n (%) 1-2x/day 1-3x/week 1-3x/month |
8 (14.8) 43 (79.6) 3 ( 5,6 ) |
2 (6.7) 19 (63.3) 9 (30) |
0.009 |
4 (11.8) 28 (82.4) 2 (5.9) |
1 (5.3) 14 (73.7) 4 (21.1) |
0.251 |
5 (11.1) 33 (73.3) 7 (15.6) |
5 (12.8) 29 (74.4) 5 ( 12,8 ) |
0.943 |
Form of fish supply n (%) boiled baked |
46 (85.2) 8 (14.8) |
21 (70) 9 (30) |
0.155 |
28 (82.4) 6 (17.6) |
14 (73.7) 5 (26.3) |
0.496 |
35 (77.8) 10 (22.2) |
32 (82.1) 7 (17.9) |
0.787 |
Frequency of tree nuts supply n (%) 1-2x/day 1-3x/week 1-3x/month |
2 (22.2) 2 (22.2) 5 (55.6) |
0 (0) 6 (50) 6 (50) |
>0.05 |
2 (25) 2 (25) 4 (50) |
0 (0) 6 (50) 6 (50) |
>0.05 |
2 (25) 2 (25) 4 (50) |
0 (0) 6 (46.2) 7 (53.8) |
>0.05 |
Form of tree nuts supply n (%) raw boiled baked |
7 (77.8) 2 (22.2) 0 (0) |
8 (66.7) 0 (0) 4 (33.3) |
>0.05 |
6 (75) 2 (25) 0 (0) |
8 (66.7) 0 (0) 4 ( 33,3 ) |
>0.05 |
6 (75) 2 (25) 0 (0) |
9 (69.2) 0 (0) 4 (30.8) |
>0.05 |
Frequency of shellfish supply n (%) 1-2x/day 1-3x/week 1-3x/month |
0 (0) 0 (0) 2 (100) |
0 (0) 0 (0) 4 (100) |
>0.05
|
0 (0) 0 (0) 2 (100) |
0 (0) 0 (0) 4 (100) |
>0.05
|
0 (0) 0 (0) 2 (100) |
0 (0) 0 (0) 4 (100) |
>0.05
|
Form of shellfish supply n (%) boiled baked |
2 (100) 0 (0) |
3 (75) 1 (25) |
1.00 |
2 (100) 0 (0) |
3 (75) 1 (25) |
1.00 |
2 (100) 0 (0) |
3 (75) 1 (25) |
1.00
|
- For milk, it will be good to know whether any of these children had received formula for the first few days of life before being subsequently exclusively breastfed.
We are aware that formula feeding in the first days of life increases the risk of developing allergies, unfortunately, we did not take this into account in our survey. The questionnaire was very extensive and it was necessary to limit the number of questions.
We acknowledge that this question would be an important and valuable addition to our study, so we will take it into account next time.
- It is not clear how they were fed but one is assuming that they were exclusively breastfed so you may want to clarify this as well.
Also whether any of these children received intermittent supplementation of formula in the first few months of life. Many of these children are firstborn infants, so it may take time for mothers to establish breastfeeding. This is more so as you have made some comments about this in the discussion.
Children enrolled in the study were fed in different ways. Only some of them were exclusively breastfed. Detailed data are presented in the Table below.
Type of feeding |
Group A+C n=55 (100%) |
Group B+D n=31 (100%) |
P |
Group A n=35 (100%) |
Group D n=20 (100%) |
P |
Group A+B (100%) |
Group C+D n=40 (100%) |
P |
Exclusively breastfeeding, n (%) no yes |
50 ( 90.9 ) 5 ( 9.1 ) |
26 ( 83.9 ) 5 ( 16.1) |
0.24 |
32 ( 97.1 ) 3 ( 2.9 ) |
17 ( 85 ) 3 ( 15 ) |
0.54 |
41 ( 89.1 ) 5 ( 10.9 ) |
35 ( 87.5) 5 ( 12.5 ) |
0.7 |
Breastfeeding time, month median (Q1 – Q3) |
5 ( 4 - 10 ) |
7 (6 - 11) |
0.07 |
5 ( 3 - 10 ) |
8 ( 6 - 12 ) |
0.04 |
6 (3 – 9.5) |
6.5 (5- 11) |
0.13 |
Exclusively formula feeding, n (%) no yes |
17 ( 30.9 ) 38 ( 69.1 ) |
16 ( 51.6 ) 15 ( 48.4 ) |
0.181 |
12 ( 34.3 ) 23 ( 65.7 ) |
8 ( 40 ) 12 ( 60 ) |
0.779 |
20 ( 43.5) 26 ( 56.5 ) |
13 ( 32.5 ) 27 ( 67.5 ) |
0.514 |
Formula feeding time, (month) median (Q1 – Q3) |
7 ( 6 - 9 ) |
6 (5 – 10.5 ) |
0.864 |
8 ( 5.5 - 9 ) |
6 ( 5 - 9 ) |
0.641 |
8 ( 5 - 9 ) |
7 ( 6 - 8 ) |
0.59 |
Mixed feeding, n(%) no yes |
43 ( 78.2) 12 ( 21.8) |
20 ( 64.5 ) 11 ( 35.5) |
0.285 |
26 ( 74.3 ) 9 ( 25.7) |
15 ( 75 ) 5 ( 25 ) |
1,00
|
31 ( 67.4 ) 15 ( 32.6 ) |
32 ( 80 ) 8 ( 20) |
0.194 |
Mixed feeding time, month median (Q1 – Q3) |
7.5 ( 6 - 9 ) |
5 ( 4 - 9 ) |
0.322 |
7.5 (6.5 - 9 ) |
4 ( 3 – 6.5 ) |
0,229 |
7 ( 6 - 9 ) |
4.5 ( 4 - 9 ) |
0.268 |
In general, studies on the comparison of breastfeeding vs formula feeding for the purpose of allergy prevention are very difficult to perform and are burdened with the possibility of an inverse cause-and-effect relationship (reverse causation). In our study, there were no statistically significant differences between the examined groups of children according to the type of feeding in the early period of life. Although we showed that the longer duration of breastfeeding reduced the risk of allergy and sensitization, the impact of breastfeeding on allergy risk evaluation was not the aim of the study. The small number of children made it impossible to draw appropriate conclusions on this subject. It is well known that exclusive breastfeeding is recommended for all mothers, although there is no specific link between exclusive breastfeeding and primary FA prevention. Moreover, there is no evidence that formula-fed infants or non-exclusively breast-fed infants are at a greater or lesser risk of developing food allergy compared with exclusively breast-fed infants.
- When you look at the graph presented, the majority of children who had fish allergy as well as those who did not, had regular intake of fish at 1-3x/week. Were these children allergic to all types of fish or only to specific fish?
The presented graph relating to the frequency of fish consumption does not apply to children who have been shown to be allergic to fish only, but to children with allergies in general. The survey did not specify the type of fish consumed.
- For peanut, what forms of raw peanuts do parents in Poland give? Most of the cohort studies (real life), the peanut is cooked/roasted.
Raw peanuts refer to peanuts in the shell that have not been roasted.
The parents used peanut flour, made from uncooked peanuts.
- In terms of food diversity, you mentioned that early introduction of a varied diet reduced the risk of food allergy. This was probably true at 6 months and 12 months but not at 3 months. Would you like to explain why this might be?
We agree with the Reviewer's opinion that a more varied diet at 6 and 12 months of age reduces the risk of allergies. The question related to the <3 month period does not refer to the diversity of the diet, but concerns the relationship between the introduction of solid foods and their impact on the development of allergies.
- In terms of risk factors for food allergy, you have identified family history of inhalation allergy. Did you look at the children who had eczema and their risk of food allergy and especially whether the severity of the eczema was associated with a higher risk?
In the conducted study, we took into account various risk factors that may affect the occurrence of allergies, which is the subject of another publication.
Majority of children with food allergies had moderate or severe AD.
I would like once again to thank you for all your help and support. I truly believe your comments have helped make this article more valuable. I hope that this manuscript will be now accepted for publication in Nutrients, which would be considered a great honor for me. Thank You for Your time and consideration.
With best regards,
Magdalena Chęsy
Aneta Krogulska

Round 2
Reviewer 2 Report
Thank you for the revised article.
The participants are probably from a slightly skewed population as they are recruited from the Allergology and Gastroenterology Services which accounts for the relatively high number of allergic and sensitized patients compared to healthy children. This can affect your conclusions together with the small number of patients recruited. Otherwise, the revised article is better presented. Just minor English edits may be needed.